# BRSSD10K : A SEGMENTATION DATASET OF BANGLADESHI ROAD SCENARIO

## ABSTRACT

In this paper, we present a novel Bangladeshi Road Scenario Segmentation Dataset designed to advance autonomous driving technologies under the challenging and diverse road conditions of Bangladesh. This comprehensive instance segmentation dataset comprised 10,082 high-resolution images captured across nine major cities, including Dhaka, Sylhet, Chittagong, and Rajshahi, addressing the critical need for region-specific computer vision data in developing countries. Unlike existing autonomous driving datasets that primarily focus on western road conditions, BRSSD10k encompasses a wide range of environments unique to Bangladesh, including unstructured urban areas, hilly terrains, village roads, and densely populated city centers. The dataset features instance segmentation annotations with classes specifically tailored to reflect the distinctive elements of Bangladeshi roads, such as rickshaws, CNGs (auto-rickshaws), informal roadside stalls, and various nonstandard vehicles. To demonstrate its utility as a benchmarking tool for autonomous driving systems, we present comparative results from several state-of-the-art instance segmentation models tested on this dataset, achieving an mAP of 0.441. This evaluation not only showcases the dataset's effectiveness in assessing model performance but also underscores the need for adaptive algorithms capable of handling diverse and unpredictable urban environments in the context of autonomous navigation.

## 1 INTRODUCTION

Autonomous driving technologies have made substantial progress in recent years, yet their development and testing remain predominantly focused on road conditions found in Western countries. This emphasis has resulted in a significant gap in resources for developing autonomous systems capable of navigating the diverse and challenging environments present in many developing nations. To address this issue, we introduce the Bangladesh Road Scenario Segmentation Dataset (BRSSD10k), a comprehensive instance segmentation dataset specifically designed to capture the unique road conditions in Bangladesh.

Existing datasets, such as Cityscapes Cordts et al. (2016) and Mapillary Vistas Neuhold et al. (2017), were created with a focus on Western locations. While these datasets have been instrumental in advancing computer vision for autonomous driving, they do not reflect the complexities of non-Western environments. The Indian Driving Dataset (IDD) Varma et al. (2018), with 10,000 annotated images, has advanced research in the subcontinent, yet even it does not fully encapsulate the intricate road scenarios found in Bangladesh. Cityscapes, with its 5,000 finely annotated images of urban scenes from German cities, remains a benchmark for structured environments, while IDD represents a step toward more diverse scenarios by capturing the heterogeneous nature of Indian roads. However, neither dataset comprehensively addresses the unique challenges posed by Bangladeshi roads, where the interaction between formal and informal transportation systems presents distinct difficulties for computer vision models.

Instance segmentation, which involves both classifying and delineating individual object instances within an image, is crucial for autonomous navigation in complex environments He et al. (2017). The dense traffic, non-motorized vehicles, and fluid road usage in Bangladeshi cities demand highly accurate and robust instance segmentation models. BRSSD10k was developed to meet these re-

quirements by offering a large-scale, finely annotated dataset that reflects the specific characteristics of Bangladeshi roads.

Our contributions are as follows:

1. We present BRSSD10k, a dataset containing 10,082 high-resolution images and 138,052 instance segmentation annotations captured across nine major cities in Bangladesh.

2. We introduce novel classes specific to the road conditions in Bangladesh, including rickshaws, CNGs (auto-rickshaws), and informal roadside stalls, enabling the development of more contextually aware autonomous systems.

3. We provide benchmark results using state-of-the-art instance segmentation models, highlighting the unique challenges of Bangladesh's road conditions and establishing a new baseline for performance in such environments.

## 2   RELATED WORKS

Table 1 presents a comparative analysis of BRSSD10k alongside three prominent datasets in autonomous driving research: Cityscapes, Mapillary Vistas, and the Indian Driving Dataset (IDD). BRSSD10k, with 10,082 images, is comparable in size to IDD and offers twice the number of images as Cityscapes, though less than Mapillary Vistas' 25,000. It matches IDD with 34 object categories, positioning itself between Cityscapes' 30 and Mapillary Vistas' extensive 124 classes. While each dataset has a unique geographic focus – Cityscapes on German urban areas, IDD on Indian cities, and Mapillary Vistas offering global coverage – BRSSD10k concentrates on nine major Bangladeshi cities, filling a crucial gap in representation of diverse urban environments in developing nations.

Table 1: Comparison of Cityscapes, Mapillary Vistas, IDD, and BRSSD10k Datasets

| Feature | Cityscapes | Mapillary Vistas | IDD | BRSSD10k |
|---|---|---|---|---|
| Number of Images | 5,000 images | 25,000 images | 10,000 images | 10,082 images |
| Object Categories | 30 classes | 124 classes | 34 classes | 34 classes |
| Geographic Coverage | Primarily urban areas in Germany | Global coverage (multiple continents) | Primarily urban areas in India | Nine major cities in Bangladesh |
| Use Cases | Urban scene understanding | Autonomous driving, semantic segmentation | Autonomous driving, scene understanding | Autonomous driving in diverse conditions |

## 3   DATASET

### 3.1   PROBLEM STATEMENT

Let $\mathcal{D} = \{(\mathbf{I}_i, \mathbf{M}_i)\}_{i=1}^N$ be a training set of $N$ labeled images $\mathbf{I}_i \in \mathcal{X}$ and their corresponding ground-truth instance segmentation masks $\mathbf{M}_i$. Each $\mathbf{M}_i$ is a set of instance masks $\{\mathbf{m}_{ij}\}_{j=1}^{K_i}$, where $K_i$ is the number of instances in image $\mathbf{I}_i$, and each $\mathbf{m}_{ij} \in \{0,1\}^{H \times W}$ represents a binary mask for the $j$-th instance in the $i$-th image, with $H$ and $W$ being the height and width of the image, respectively.

The task of instance segmentation is to learn a model $f_{\boldsymbol{\theta}} : \mathcal{X} \to \mathcal{Y}$, where $\boldsymbol{\theta}$ is a set of learnable parameters. In this context, $\mathcal{Y}$ represents the set of instance segmentation masks for the detected objects, along with their corresponding class labels and confidence scores.

Given a test image $\mathbf{I}$ from the diverse road scenarios of Bangladesh, the trained model predicts a set of instance masks $\mathbf{M}_p = \{\mathbf{m}_{pk}\}_{k=1}^K$, where $K$ is the number of detected instances. Each predicted mask $\mathbf{m}_{pk} \in [0,1]^{H \times W}$ is accompanied by a class label $c_k \in \mathcal{C}$, where $\mathcal{C}$ is the set of predefined classes specific to Bangladeshi road scenes (e.g., cars, rickshaws, pedestrians, roadside stalls), and a confidence score $s_k \in [0,1]$.

### 3.2   CHALLENGES OF BANGLADESHI DATASETS

The complexity of Bangladeshi roads presents significant challenges for traffic modeling and analysis, driven by a combination of ambiguous boundaries, diverse vehicle types, unpredictable pedestrian behavior, and varied environmental conditions. Unlike the clearly defined road edges seen

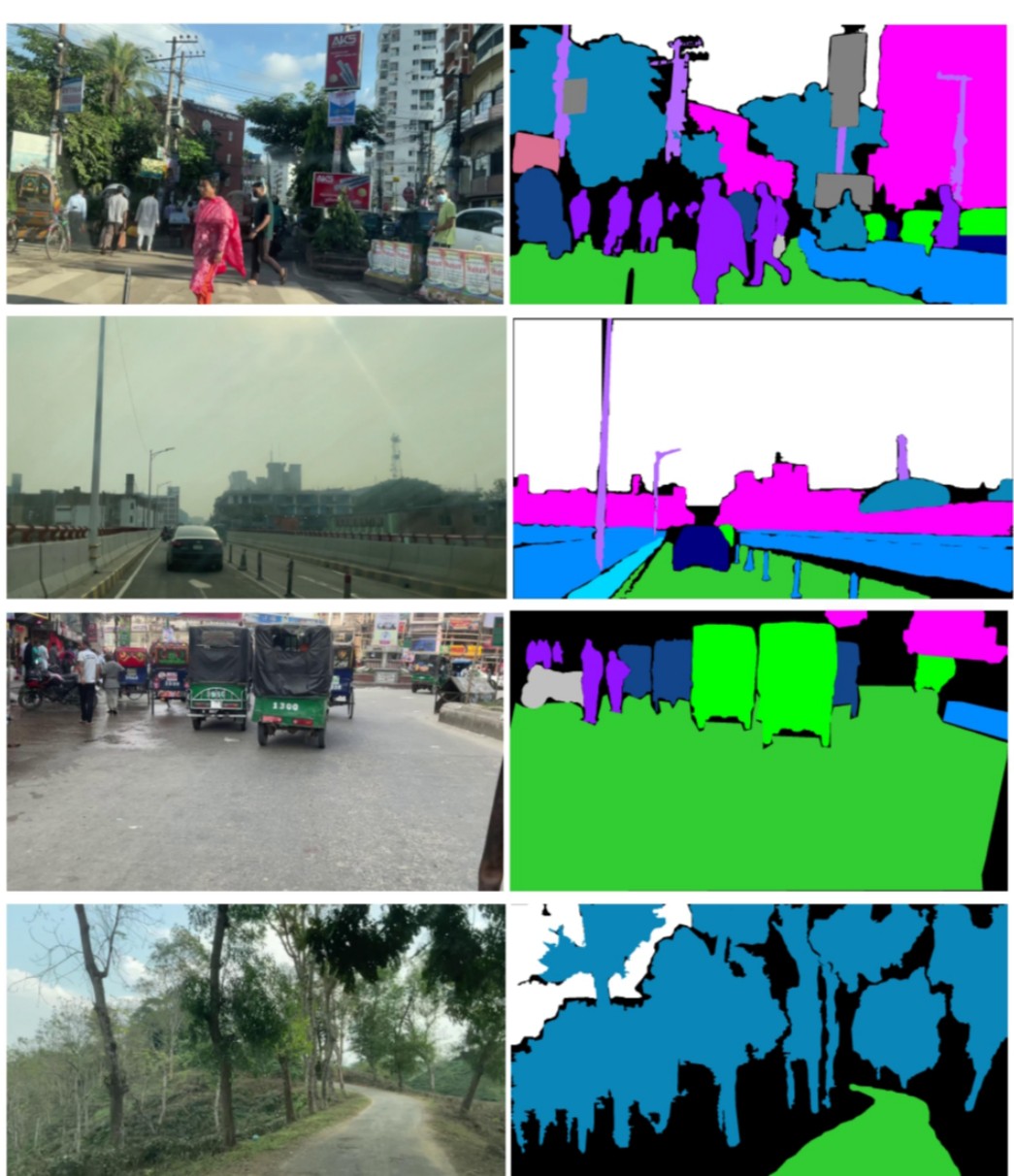

Figure 1: Sample Images from BRSSD10k with Masked Annotations

in datasets such as Cityscapes, Bangladeshi roads often transition seamlessly into unpaved areas, which may be drivable in some instances. This ambiguity often results in misclassifications by models trained on more structured datasets, leading to potential safety risks.

Moreover, the roadways are teeming with a wide variety of vehicles that reflect the local transport culture. In addition to traditional cars and trucks, the streets are filled with rickshaws, CNGs (compressed natural gas auto-rickshaws), and modified local vehicles such as 'Lagunas' and 'Nosimons.' These unique vehicles operate differently from standard vehicles, exhibiting variations in speed, maneuverability, and compliance with traffic regulations. This diversity extends to the conditions of the vehicles themselves, which often show signs of wear and tear and include many older models, contributing to the complexities of traffic interactions.

Pedestrian behavior in Bangladesh further complicates road dynamics. Individuals frequently cross streets at arbitrary locations rather than using designated crosswalks, increasing the potential for

conflicts between vehicles and pedestrians. Additionally, many road users, including rickshaws, CNGs, and motorcycles, often disregard traffic rules, leading to unpredictable traffic patterns and a lack of correlation with road signage, such as lane markings and traffic lights.

The presence of extensive information boards, including billboards and shop signs, adds another layer of complexity. These displays, especially in urban areas, provide valuable context for localization and mapping efforts, often highlighting landmarks or indicating nearby buildings. However, they can also create visual clutter that may confuse both human drivers and automated systems.

Moreover, the terrain in certain regions of Bangladesh, such as the hill tracts, introduces additional challenges. Roads in these areas can be narrow and winding, with steep gradients and sharp turns that require specialized navigation skills. The lack of well-defined road boundaries in these hill tracks, combined with unpredictable weather conditions and limited visibility, makes driving even more difficult. The unique geographical features of these regions necessitate careful consideration in traffic modeling to accommodate the specific behaviors of both vehicles and pedestrians in these environments.

We can see in Figure 1, the diversity and complexity of road environments in Bangladesh as captured by the BRSSD10k dataset. The image includes four distinct road scenarios, each paired with its corresponding segmentation map. These scenarios featured busy urban streets in cities, rural village roads, expressways, and hill tracks. Each pair of images, an original photo and its segmentation map, demonstrates the dataset's ability to accurately label and distinguish various road users, vehicles, infrastructure, and natural features unique to Bangladesh. The segmentation maps provide detailed annotations of objects, such as pedestrians, vehicles, buildings, and vegetation, showcasing high-quality labeling within the dataset. This visual representation highlights the comprehensive coverage of different road types in Bangladesh, from dense city streets to remote hilly tracks and expressways. The BRSSD10k dataset offers valuable resources for developing computer vision models capable of navigating the diverse and complex traffic conditions found in these varied environments.

## 4 DATA ACQUISITION AND LABELING

The Bangladesh Road Scenario Segmentation Dataset (BRSSD10k) was compiled through a rigorous process of data collection, preprocessing, and annotation. Our methodology ensured the capture of authentic and diverse road scenarios specific to Bangladesh, while maintaining high-quality annotations.

### 4.1 DATA COLLECTION

We collected raw data exclusively using smartphone cameras to capture real-world road scenarios across Bangladesh. This approach allowed us to gather a wide range of urban and rural road scenes, reflecting the true diversity and challenges of the country's transportation infrastructure. Importantly, no images were sourced from online platforms, ensuring the dataset's originality and relevance to the specific context of Bangladesh.

BRSSD10k includes data from nine key locations: Dhaka, Sherpur, Mymensingh, Khulna, Sylhet, Maowa, Juri, Rajshahi, and Chittagong. These locations were strategically chosen to represent the country's diverse road conditions, covering major urban centers like Dhaka and Chittagong, regional hubs such as Khulna and Sylhet, smaller towns like Sherpur and Juri, and areas with unique geographic features like Maowa. This geographic variety ensures that the dataset reflects the full spectrum of road scenarios in Bangladesh, including both congested city streets and rural roads.

### 4.2 PREPROCESSING

The collected videos were preprocessed to extract individual frames at a rate of one frame per second. This extraction rate strikes a balance between capturing temporal variations and maintaining a manageable dataset size. Each extracted frame was standardized to a resolution of 1280x720 pixels, ensuring sufficient detail for complex scene analysis while considering computational efficiency for future model training. Additionally, some frames were extracted at a resolution of 848x478 pixels.

### 4.3 ANNOTATION PROCESS

The annotation process was carried out on the Roboflow platform, chosen for its robust features and collaborative capabilities. Our annotation team consisted of 10 trained annotators who were familiar with the local context and the specific requirements of our dataset.

### 4.4 QUALITY ASSURANCE

To ensure the highest possible annotation accuracy, we implemented a two-stage validation process:

1. Initial Annotation: Each image was manually annotated by one of the 10 trained annotators.
2. Validation: Following the initial annotation, each image underwent a secondary review by two different individuals. This dual-validation approach helped in identifying and correcting any potential errors or inconsistencies in the annotations.

This meticulous process of data acquisition, preprocessing, and multi-stage annotation validation was designed to minimize errors and ensure the reliability of our dataset. The resulting BRSSD10k dataset provides a high-quality, context-specific resource for advancing autonomous driving research and development in Bangladesh and similar developing countries.

## 5 DATASET STATISTICS

Figure 2: Class distrubution of BRSSD10k Dataaset

### 5.1 CLASS DISTRIBUTION ANALYSIS

Figure 2 exhibits a diverse and imbalanced class distribution, reflecting the complexity of urban Bangladeshi road scenes. Person instances (22,357) dominate the dataset, followed closely by vegetation (17,659), highlighting the densely populated and green urban environments. Road infrastructure elements such as roads (12,419) and poles (14,174) are well-represented. Notably, auto-rickshaws (12,937) and three-wheelers (8,795) have high instance counts, underscoring their prevalence in Bangladeshi traffic. However, the dataset shows significant class imbalance, with critical but less frequent objects like traffic lights (38), construction vehicles (28), and road blockers (13)

being underrepresented. This imbalance poses challenges for model training and emphasizes the need for specialized data augmentation or balancing techniques to ensure robust detection across all classes, particularly for safety-critical objects in autonomous driving applications.

## 5.2 LOCATION WISE IMAGE DISTRIBUTION

Table 2 presents the geographical distribution of images in our dataset across various locations in Bangladesh. The dataset comprises a total of 10,082 images collected from nine distinct regions. Khulna contributes the largest portion with 3,011 images, followed by Sylhet (1,508) and Juri (1,244). Maowa, Dhaka, and Mymensingh provide 1,020, 930, and 897 images respectively. Sherpur accounts for 741 images, while Chittagong contributes 563. Rajshahi has the smallest representation with 168 images. This diverse geographical spread enhances the dataset's ability to capture regional variations, potentially improving the robustness and generalizability of models trained on this data.

Table 2: Location-Wise Image Counts

| LOCATION | COUNT |
| --- | --- |
| Dhaka | 930 |
| Sherpur | 741 |
| Mymensingh | 897 |
| Khulna | 3011 |
| Sylhet | 1508 |
| Maowa | 1020 |
| Juri | 1244 |
| Rajshahi | 168 |
| Chittagong | 563 |

## 6 DATASET CLASS DEFINITION

BRSSD10k introduces a novel class definition system tailored to Bangladesh's unique road environments. Our approach balances comprehensiveness with practicality, addressing the specific challenges of autonomous driving in this region.

### 6.1 VEHICLE CLASSES

We adopt the vehicle classification from the BadODD dataset Baig et al. (2024), chosen for its scalability and relevance to Bangladesh's diverse vehicle types. This system efficiently categorizes the wide range of motorized and non-motorized vehicles prevalent on Bangladeshi roads.

### 6.2 ROAD ENVIRONMENT CLASSES

To capture the complexity of local road scenarios, we introduce several key classes:

- **Road:** Primary driving surface.
- **Road_sign:** Traffic and informational signage.
- **Road_divider:** Includes roadside and median dividers, and temporary barriers.
- **Road_blocker:** Obstacles or intentional road blockades.
- **Speed_breaker:** Common speed control structures.
- **Toll:** Identifies toll plazas for navigation through checkpoints.
- **Rail_crossing:** Critical for safety at railway intersections.
- **Garbage_bin:** Often encroaching on urban road space.
- **Poster:** Suspended advertisements that may obstruct passage.

- **Wall and Gate:** Important for identifying building entrances.
- **Fence:** Common in rural areas, delineating boundaries.

### 6.3 ADDITIONAL ENVIRONMENTAL CLASSES

We further enhance the dataset's utility with classes such as:

- **Animal:** Annotation of livestock commonly encountered on roads.
- **Pole, Overbridge, Billboard:** Key urban infrastructure elements.
- **Sidewalk:** Pedestrian pathways.
- **Sky:** For horizon detection and scene understanding.
- **Traffic_light:** Essential for traffic management.
- **Vegetation:** Affects road visibility and navigation.

This class system is designed to capture the full spectrum of elements in Bangladesh's complex road scenarios. Notable inclusions like rail crossings, garbage bins, and animals reflect real-world challenges often overlooked in datasets from more developed regions.

The **Road_sign** class, for instance, enables future integration with OCR technologies, potentially allowing autonomous systems to interpret and act on signage information in real-time. Similarly, the detailed categorization of road dividers and blockers addresses the fluid nature of traffic management in many Bangladeshi urban areas.

By providing such a comprehensive yet locally relevant classification, BRSSD10k offers a robust foundation for developing autonomous driving systems capable of navigating Bangladesh's unique road environments. This approach not only enhances the dataset's immediate applicability but also contributes valuable insights to the broader field of autonomous driving research, particularly in diverse and challenging road conditions.

## 7 MODEL TRAINING

### 7.1 DATASET SPLIT

The BRSSD10k dataset is divided into three subsets to support effective training and evaluation of models for autonomous driving technologies, as detailed in Table 3. The training set consists of 6,020 images, enabling robust model development by providing a comprehensive range of road scenarios. The validation set, comprising 2,018 images, facilitates the fine-tuning of model parameters and selection of optimal configurations to enhance generalization capabilities. Lastly, the test set, with 2,044 images, serves as an unbiased benchmark for assessing model performance on unseen data, ensuring rigorous evaluation.

Table 3: BRSSD10k Dataset Split

| Split | Number of Images |
| --- | --- |
| Train | 6,020 |
| Validation | 2,018 |
| Test | 2,044 |

### 7.2 MODELS

In this study, we evaluate the performance of four state-of-the-art object detection models on our BRSSD10k dataset: YOLOv5 Jocher (2020), YOLOv8 Jocher et al. (2023) and YOLOv9 Wang et al. (2024). Each model represents a different approach to object detection and instance segmentation, allowing us to comprehensively assess their capabilities in the context of Bangladesh's complex road scenarios.

## 7.3 YOLOv5

YOLOv5 is an improvement over previous YOLO versions, offering enhanced speed and accuracy. It utilizes a CSPNet backbone and PANet neck for feature extraction and aggregation, respectively, making it highly efficient for real-time object detection.

**Loss Function:** YOLOv5 employs a composite loss function consisting of three components:

$$L_{total} = \lambda_{coord}L_{box} + \lambda_{obj}L_{obj} + \lambda_{class}L_{class} \tag{1}$$

where $L_{box}$ is the bounding box regression loss (typically a combination of MSE and IoU loss), $L_{obj}$ is the objectness loss, and $L_{class}$ is the classification loss (typically cross-entropy).

## 7.4 YOLOv8

YOLOv8 further refines the YOLO architecture, introducing improvements in both speed and accuracy. It incorporates a more sophisticated backbone and neck structure, and introduces anchor-free detection heads for better performance.

**Loss Function:** YOLOv8 uses a similar composite loss function to YOLOv5, but with refined components:

$$L_{total} = \lambda_{box}L_{box} + \lambda_{cls}L_{cls} + \lambda_{dfl}L_{dfl} \tag{2}$$

where $L_{box}$ is the bounding box regression loss, $L_{cls}$ is the classification loss, and $L_{dfl}$ is the distribution focal loss for better localization.

## 7.5 YOLOv9

YOLOv9 represents the latest iteration in the YOLO family, introducing novel concepts such as programmable gradient information and implicit knowledge learning . These innovations aim to enhance the model's ability to generalize and perform well on diverse datasets.

**Loss Function:** YOLOv9's loss function builds upon YOLOv8's, with additional components to account for its new features:

$$L_{total} = \lambda_{box}L_{box} + \lambda_{cls}L_{cls} + \lambda_{dfl}L_{dfl} + \lambda_{aux}L_{aux} \tag{3}$$

where $L_{aux}$ represents auxiliary losses that help in training the implicit knowledge components.

## 7.6 HYPERPARAMETERS

The hyperparameter configurations for training the YOLOv5, YOLOv8, and YOLOv9 models are detailed in Tables 4 and 5, outlining the essential training parameters. Both YOLOv5 and YOLOv8 were trained for 100 epochs with a batch size of 16, using the AdamW optimizer and a learning rate of 0.001. In contrast, the YOLOv9 model was specifically trained with a batch size of 2 to fit within the memory constraints of the NVIDIA RTX 4080 SUPER, which has 16 GB of VRAM. This adjustment in batch size was necessary to accommodate the model's requirements without exceeding the available VRAM. The consistent use of the same optimizer and learning rate across the models facilitates comparative analysis of their performance, while the powerful GPU setup enables efficient handling of complex datasets, enhancing the models' capabilities in segmentation tasks.

Table 4: Hyperparameter configuration for YOLOv5 and YOLOv8 training

| HYPERPARAMETERS | VALUES |
|---|---|
| Epoch | 100 |
| Batch Size | 16 |
| Optimizer | AdamW |
| Learning Rate (LR) | 0.001 |

Table 5: Hyperparameter configuration for YOLOv9 training

| HYPERPARAMETERS | VALUES |
|---|---|
| Epoch | 100 |
| Batch Size | 2 |
| Optimizer | AdamW |
| Learning Rate (LR) | 0.001 |

## 8 RESULT AND DISCUSSION

Table 6 presents a comparative analysis of mean Average Precision (mAP) scores at 50% Intersection over Union (IoU) threshold for three versions of the YOLO (You Only Look Once) object detection algorithm. The table delineates the performance metrics for YOLOv5, YOLOv8, and YOLOv9 across both validation and test datasets. Notably, YOLOv8 demonstrates superior performance, achieving the highest mAP50 scores of 0.404 and 0.441 on the validation and test sets, respectively. YOLOv9 follows closely in validation performance with a mAP50 of 0.406, but shows a slight decrease in test set performance with a mAP50 of 0.419. YOLOv5, while still competitive, exhibits lower mAP50 scores of 0.339 and 0.376 for validation and test sets, respectively. These results underscore the incremental improvements in object detection capabilities across successive YOLO iterations, with YOLOv8 emerging as the most effective variant in this comparative study.

Table 6: Comparison of mAP50 Scores for Different YOLO Versions

| YOLO Version | val mAP50 | test mAP50 |
|---|---|---|
| YOLOv5 | 0.339 | 0.376 |
| YOLOv8 | 0.404 | 0.441 |
| YOLOv9 | 0.406 | 0.419 |

Figure 3 presents a comprehensive visual comparison of object detection performance across YOLOv5, YOLOv8, and YOLOv9 models on diverse traffic scenes. The figure is structured in a grid format, showcasing five distinct scenarios, each represented by a row of images. For each scenario, the original source image is displayed alongside its corresponding ground truth annotations and the detection results from the three YOLO versions. This juxtaposition allows for a nuanced analysis of each model's capabilities in identifying and localizing various objects such as vehicles, pedestrians, and road infrastructure. Notably, the progression from YOLOv5 to YOLOv9 demonstrates incremental improvements in detection accuracy and confidence, as evidenced by the more precise bounding boxes and higher confidence scores in the later versions. The color-coded overlays in the detection results provide immediate visual cues to the models' performance, with variations in object classification and segmentation clearly visible across the different YOLO iterations. This comparative visualization effectively illustrates the evolution of YOLO architectures and their enhanced ability to handle complex, real-world traffic scenarios with increasing sophistication.

## 9 CONCLUSION

The Bangladesh Road Scenario Segmentation Dataset (BRSSD10k) represents a significant step forward in addressing the unique challenges of autonomous driving in diverse and complex urban environments. By providing a comprehensive, finely annotated dataset specific to Bangladesh's road conditions, BRSSD10k fills a critical gap in the existing landscape of autonomous driving datasets.

Our work demonstrates the importance of region-specific data in developing robust and adaptable computer vision models for autonomous navigation. The inclusion of novel classes tailored to Bangladesh's road scenarios, such as rickshaws, CNGs (auto-rickshaws), and informal roadside structures, enables more accurate and culturally aware autonomous systems. Furthermore, the benchmark results presented highlight the unique challenges posed by Bangladesh's road condi-

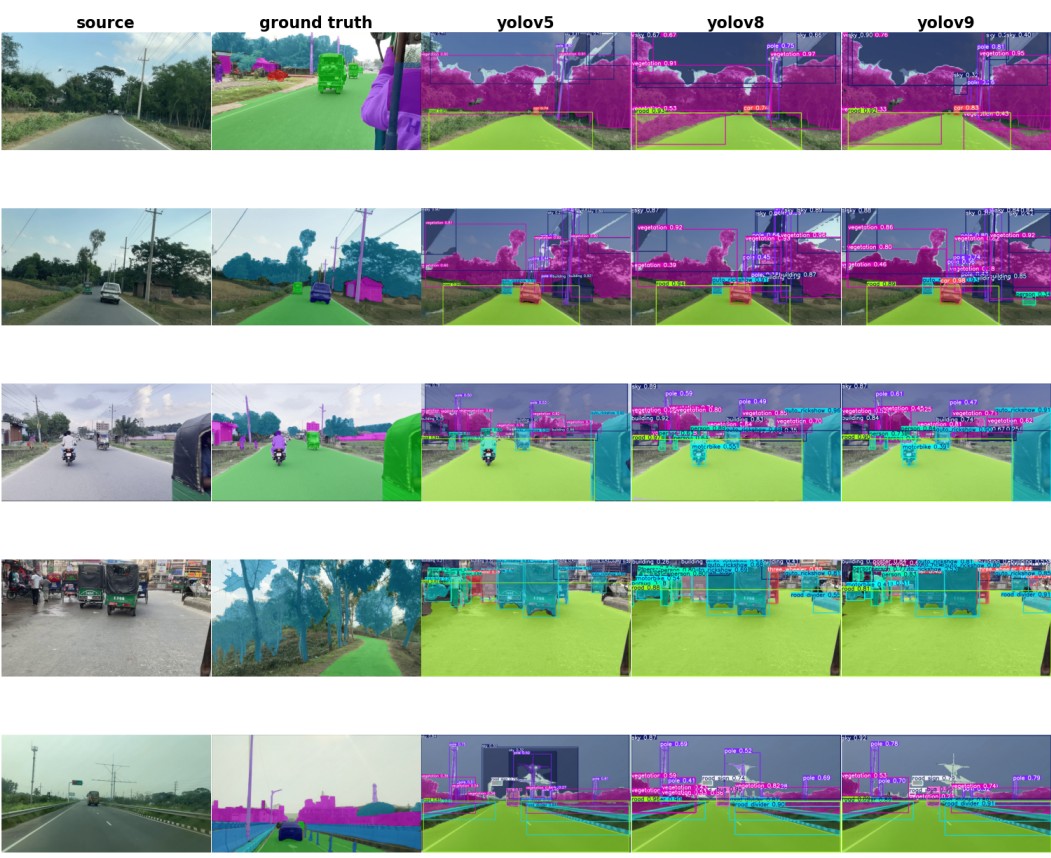

Figure 3: Predictions of YOLOv5, YOLOv8 and YOLOv9 models

tions and set a new baseline for performance in these environments. However, the limitations of the dataset are discussed below:

1. Lack of nighttime imagery: BRSSD10k currently does not include images captured during nighttime conditions, which represent a significant aspect of real-world driving scenarios.

2. Absence of adverse weather conditions: The dataset does not encompass images from rainy conditions or muddy road surfaces, which are common during Bangladesh's monsoon season and can significantly impact driving conditions.

3. Limited road surface variations: While the dataset covers a wide range of urban and rural scenes, it does not extensively capture extremely challenging road surfaces that may be encountered in more remote areas.

Additionally, to provide a more robust evaluation of the dataset's effectiveness, future work should include benchmarking against state-of-the-art Vision Language Models (VLMs). This comparison would offer valuable insights into the dataset's performance relative to more generalized models and highlight areas where region-specific data provides significant advantages.

Despite these limitations, BRSSD10k represents a valuable contribution to the field of autonomous driving research. By focusing on the unique challenges presented by Bangladesh's road conditions, this dataset not only advances the development of autonomous technologies for similar environments but also broadens the global understanding of diverse driving scenarios. As autonomous driving research continues to evolve, datasets like BRSSD10k will play a crucial role in creating more inclusive and adaptable systems capable of operating safely and efficiently in a wide range of global contexts.

**Reproducibility Statement** To facilitate the reproducibility of our results, we have provided all the hyperparameter configuration in the paper. Additionally, a comprehensive package containing our training and inference notebooks, along with detailed instructions for their use. This package is available as a compressed file, which includes sample images for testing purposes. The notebooks are accompanied by information about our system specifications to ensure transparency regarding the computational environment used in our experiments. Link to the file: https://drive.google.com/file/d/1qeD3h2CzN9C6IshsVydGVbBVGPummsTF/view?usp=sharing

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
