# OpenReview forum: "BRSSD10k :  A SEGMENTATION DATASET \\OF BANGLADESHI ROAD SCENARIO"
_ICLR.cc/2025/Conference — Submitted to ICLR 2025_

### Official Review · Reviewer_zfNC · 2024-10-26

**Soundness:** 1
**Presentation:** 1
**Contribution:** 2
**Rating:** 1
**Confidence:** 4

**Summary:**

This paper proposes a road  segmentation dataset for autonomous driving purpose.  It focuses on the scenarios in Bangladeshi and make specific adaptions in class definition and labeling. Validation experiments are conducted.

**Strengths:**

It clearly analyzes the practical scenario characteristics in Bangladeshi. The class definition and labeling process fully fits the scenarios. This dataset acts as a valuable resource for developing autonomous driving models in this country. It may also contribute to general vision perception tasks.

**Weaknesses:**

1.The authors just use one paragraph to summarize the related datasets without any detailed comparison. I do not think the authors really understand the development of this field as there are only eight references.
2. The scenarios in the dataset are more likely to be corner cases comparing with the mainstream segmentation datasets. Its universality cannot be verified.
3.The structure of the manuscript is poorly organized. The logic between sections 3-6 are chaotic.
4.It is really confusing that the authors validate the segmentation dataset with YOLO.
5. It is really funny that the GT maps in Figure 3 are wrong.

**Questions:**

The authors strongly emphasize that the main motivation of this work is that there lack segmentation datasets in Bangladeshi. It should be clarified that the contribution of a dataset does not lay in its location, but the data quality, diversity, and scale.

---

### Official Review · Reviewer_ypst · 2024-10-31

**Soundness:** 3
**Presentation:** 3
**Contribution:** 2
**Rating:** 5
**Confidence:** 4

**Summary:**

This paper presents a new dataset focused on object detection and segmentation,
tailored to the specific driving conditions in Bangladesh in terms of its
appearance and taxonomy.

The dataset encompasses ~10k camera images collected in Bangladesh using a cell
phone. The images are sourced from video chunks originating from diverse
regions, and contiguous footage is sampled to 1 Hz. The frames are annotated
with object bounding boxes and segmentation masks.

The paper motivates the dataset as helpful in developing computer vision
algorithms specific to Bangladeshi driving scenes and performs a brief
comparative analysis of different YOLO-based models trained on this dataset. The
paper helpfuly provides metadata like class and geographic distribution
histograms as well as many qualitative examples in order to help the reader get
a sense of the dataset.

While it is definitely important to promote datasets which cover a diverse range
of environments, I think the quantitative argument made in this paper to
motivate the dataset could be strengthened. For example, the argument could be
improved by showing experimental results which demonstrate the limitations of
other dataset on data collected in Bangladesh.

**Strengths:**

- [S1] Diverse object classes from multiple cities in Bangladesh, reflecting a
  unique label distribution that is materially different from other established
  datasets such as Waymo Open and nuScenes.
- [S2] The authors also present the results of a few detection baselines based
  on YOLO, trained and evaluated on this dataset's corresponding splits.

**Weaknesses:**

- [W1] The related section could be made a bit more comprehensive. For example,
  it would be interesting to also discuss other datasets focusing on non-Western
  streets, such as the dataset introduced in [@traphic]. Even though it's
  mentioned later in the paper, the BadODD dataset should also be covered in the
  related work section and in the relevant tables.
- [W2] While it is helpful to benchmark a few existing models on the proposed
  dataset, it would be beneficial to also compare these numbers with those from
  models trained on a mainstream dataset such as CityScapes or Mapillary Vistas.
  If models trained on a dataset like Cityscapes or Mapillary Vistas fail to
  perform well on this dataset, that would make for a good quantitative argument
  for why this dataset will help the community.
  - As a side-note, even if the taxonomy another dataset won't match the one in
    BRSSD10k perfectly, this gap could be alleviated by the use of an
    off-the-shelf VLM, which have been shown to be very good at tasks like open
    set object detection---see, for example, Grounding DINO [@liu2024grounding].
- Minor Suggestions
  - Sections 7.3, 7.4, and 7.5 can be shortened and replaced with more
    comparisons, or additional details about the dataset or its software
    development kit. Readers can refer to the corresponding references if they
    are curious about the specific loss functions used to train these models.
  - The citation markers seem to be missing parentheses around them. For
    example, a sentence like "... complex environments He et al. (2017)" should
    be formatted like "... complex environments (He et al., 2017)."
- References:
  - [@traphic]: Chandra, Rohan, et al. "Traphic: Trajectory prediction in dense
    and heterogeneous traffic using weighted interactions." CVPR. 2019.
  - [@liu2024grounding]: Liu, Shilong, et al. "Grounding dino: Marrying dino
    with grounded pre-training for open-set object detection." arXiv preprint
    arXiv:2303.05499 (2023).

**Questions:**

- [Q1] How is the dataset split into train/val/test? Do you perform geographic
  splitting, or is the splitting purely at random?

---

### Official Review · Reviewer_bfsm · 2024-11-01

**Soundness:** 3
**Presentation:** 3
**Contribution:** 2
**Rating:** 3
**Confidence:** 5

**Summary:**

The manuscripts presents a novel road-driving dataset for instance segmentation. The dataset includes more than 10000 high resolution images acquired along 9 cities in Bangladesh. The dataset taxonomy includes 34 classes that reflect typical needs of autonomous driving and regional characteristics. The taxonomy is mostly well-balanced (Figure 2). There are around 6000 training, 2000 validation and 2000 test images. The presented experiments involve object detection with stock models and report mAP50 performance on validation and test datasets.

**Strengths:**

- The dataset will likely prove as a valuable contribution to the field.

- Many stuff classes are annotated (sky, road, wall, fence).

- Little effort is required to extend the dataset for panoptic segmentation.

**Weaknesses:**

- it is hard to recommend n+1-th road-driving dataset for publication at a major conference

- dataset focuses on typical images, for which our models are known to work well

- the baseline models address only object detection (some universal segmentation model such as MaskFormer would be a better choice)

**Questions:**

It would make sense to extend the dataset with full panoptic labels.

It would make sense to cite and discuss related road driving datasets: ACDC, WildDash, FishyScapes, SegmentMeIfYouCan.

---

### Official Review · Reviewer_pH3o · 2024-11-05

**Soundness:** 1
**Presentation:** 1
**Contribution:** 1
**Rating:** 3
**Confidence:** 5

**Summary:**

The paper introduces BRSSD10k, a segmentation dataset specifically tailored to the unique and diverse road scenarios in Bangladesh. This dataset consists of 10,082 high-resolution images from nine cities across the country, with detailed annotations covering 34 classes that reflect the region's distinct transportation environment. Classes include locally prevalent elements such as rickshaws, CNGs (auto-rickshaws), and informal roadside stalls, which are critical for developing robust autonomous driving systems for Bangladeshi roads.

**Strengths:**

- The authors have compiled a comprehensive dataset with over 10,000 high-resolution images and detailed instance segmentation annotations, covering a diverse range of geographic regions within Bangladesh.

- A rigorous two-stage validation process for annotations ensures high-quality data, which is essential for developing robust and accurate computer vision models.

- Comparative evaluation with multiple state-of-the-art models (e.g., YOLOv5, YOLOv8, YOLOv9) showcases the benchmark's effectiveness and sets a baseline for future research on BRSSD10k.

- The inclusion of region-specific object classes (e.g., rickshaws, CNGs, informal stalls) provides a unique contribution, enabling autonomous systems to better understand and navigate environments outside of structured Western road layouts.

**Weaknesses:**

- The dataset only covers limited regions in one country, which is not enough to evaluate the generalization ability of segmentation.

- The quality of the segmentation masks is not satisfactory.

- Certain critical classes, such as traffic lights, construction vehicles, and road blockers, are underrepresented in the dataset.

- The dataset currently lacks nighttime and adverse weather imagery (e.g., rain or fog), which are essential for real-world segmentation.

- The paper only evaluates three versions of the YOLO model, which may limit insights into how BRSSD10k performs across different model architectures.

- There is no analysis on how models trained on BRSSD10k generalize to other datasets or vice versa.

**Questions:**

- The authors need to consider including more baselines for evaluation.

**Details Of Ethics Concerns:**

Human faces appear on the road. They are not removed and blurred.

---

### Meta-Review · Area_Chair_3fKA · 2024-12-23

**Metareview:**

The submission presents a dataset for instance segmentation on road scenes in a novel geography. While the problem is an important one, the dataset can improve in several ways. The mask qualities are quite coarse and should match state-of-the-art datasets for road scenes. The class balance might not be appropriate for semantic segmentation given under-representation for several classes. Besides instance segmentation, there could be support for panoptic or universal segmentation too, which are important in practical applications.

**Additional Comments On Reviewer Discussion:**

All reviewers recommend rejection based on the above concerns and no author rebuttal is submitted. It is suggested for the authors to improve the submission based on the numerous review suggestions. The AC agrees with the reviewer consensus that the paper may not be accepted at ICLR.

---

### Decision · Program_Chairs · 2025-01-22

Reject